# Impact of Cytochrome Induction or Inhibition on the Plasma and Brain Kinetics of [^11^C]metoclopramide, a PET Probe for P-Glycoprotein Function at the Blood-Brain Barrier

**DOI:** 10.3390/pharmaceutics14122650

**Published:** 2022-11-30

**Authors:** Louise Breuil, Nora Ziani, Sarah Leterrier, Gaëlle Hugon, Fabien Caillé, Viviane Bouilleret, Charles Truillet, Maud Goislard, Myriam El Biali, Martin Bauer, Oliver Langer, Sébastien Goutal, Nicolas Tournier

**Affiliations:** 1Laboratoire d’Imagerie Biomédicale Multimodale (BIOMAPS), Université Paris-Saclay, CEA, CNRS, Inserm, Service Hospitalier Frédéric Joliot, 4 Place du Général Leclerc, 91401 Orsay, France; 2Neurophysiology and Epileptology Department, Bicêtre Hospital, AP-HP, University Paris-Saclay, 94270 Le Kremlin-Bicêtre, France; 3Department of Clinical Pharmacology, Medical University of Vienna, 1090 Vienna, Austria; 4Department of Biomedical Imaging and Image-Guided Therapy, Medical University of Vienna, 1090 Vienna, Austria

**Keywords:** ATP-binding cassette, drug–drug interaction, membrane transporter, neuropharmacology, pharmacokinetics

## Abstract

[^11^C]metoclopramide PET imaging provides a sensitive and translational tool to explore P-glycoprotein (P-gp) function at the blood-brain barrier (BBB). Patients with neurological diseases are often treated with cytochrome (CYP) modulators which may impact the plasma and brain kinetics of [^11^C]metoclopramide. The impact of the CYP inducer carbamazepine or the CYP inhibitor ritonavir on the brain and plasma kinetics of [^11^C]metoclopramide was investigated in rats. Data obtained in a control group were compared with groups that were either orally pretreated with carbamazepine (45 mg/kg twice a day for 7 days before PET) or ritonavir (20 mg/kg, 3 h before PET) (*n* = 4 per condition). Kinetic modelling was performed to estimate the brain penetration (*V*_T_) of [^11^C]metoclopramide. CYP induction or inhibition had negligible impact on the plasma kinetics and metabolism of [^11^C]metoclopramide. Moreover, carbamazepine neither impacted the brain kinetics nor *V*_T_ of [^11^C]metoclopramide (*p* > 0.05). However, ritonavir significantly increased *V*_T_ (*p* < 0.001), apparently behaving as an inhibitor of P-gp at the BBB. Our data suggest that treatment with potent CYP inducers such as carbamazepine does not bias the estimation of P-gp function at the BBB with [^11^C]metoclopramide PET. This supports further use of [^11^C]metoclopramide for studies in animals and patients treated with CYP inducers.

## 1. Introduction

P-glycoprotein (P-gp, ABCB1) is the most studied efflux transporter expressed at the blood-brain barrier (BBB). P-gp is well-known to restrict the brain permeation of many xenobiotics, thus protecting the brain from potentially harmful substances and contributing to the sanctuary-site property of the brain [1]. P-gp may also modulate the brain distribution of many central nervous system (CNS)-active drugs, thus controlling the magnitude of their CNS effects [2,3]. Several endogenous compounds were shown to be transported substrates of P-gp, suggesting a physiological role whose importance is increasingly considered in the development of brain disorders [4]. As a consequence, P-gp function may be considered as a biomarker of BBB function which, in addition to the integrity of the BBB, may improve our understanding of the importance of the blood–brain interface in health and disease [5].

Translational methods are needed to investigate the function of P-gp at the BBB in animals and humans. Against this background, positron emission tomography (PET) imaging using radiolabeled substrates of P-gp proved to be a minimally-invasive and effective method to measure P-gp function in the living brain. PET imaging proved useful to validate protocols to block P-gp function at the human BBB using pharmacological inhibitors such as cyclosporin A or tariquidar [6]. PET imaging using [^11^C]verapamil as a P-gp substrate has been used to explore impaired P-gp function in aging [7,8,9] or in neurological disorders such as Alzheimer’s disease [10]. A [^11^C]verapamil PET protocol in patients suffering from drug-resistant epilepsy showed attenuated response to tariquidar administration in the epileptogenic focus as compared with control subjects, suggesting locally enhanced P-gp function. This protocol required two consecutive PET acquisitions (without and with tariquidar administration) in the same subject [11].

Among available PET radioligands, [^11^C]metoclopramide benefits from optimal kinetic properties for quantitative determination of P-gp function at the BBB. [^11^C]metoclopramide is considered a “weak” substrate of P-gp, which means it shows substantial brain PET signal under conditions when P-gp is fully functional at the BBB [12,13]. As a consequence, [^11^C]metoclopramide PET imaging can be used to detect either a decrease [14,15] or an increase [16] in P-gp function at the BBB using a single PET acquisition. [^11^C]metoclopramide was shown to be selectively transported by P-gp over breast cancer resistance protein (BCRP, ABCG2), another important efflux transporter at the BBB [12]. Binding of [^11^C]metoclopramide to brain parenchyma was not displaceable by co-injection of an excess of unlabeled metoclopramide, suggesting the absence of specific binding to any unintended CNS target and confirming the specificity of the PET signal for P-gp function [12,15]. Moreover, compared with other radiolabeled P-gp substrates, [^11^C]metoclopramide was shown to be more sensitive to detect small changes in P-gp function associated with partial pharmacological P-gp inhibition [14,17]. This suggests that [^11^C]metoclopramide may provide a sensitive tool to explore the dysregulation of P-gp function which is suspected to occur in many pathophysiological conditions [18].

Similar to many other PET tracers, [^11^C]metoclopramide is metabolized into radiometabolite(s) whose contribution to the PET signal is difficult to distinguish from parent (unmetabolized) radiotracer in tissues [6,19]. The chemical identity of the radiometabolites of [^11^C]metoclopramide is not known. However, radio-HPLC analysis of rat brain and plasma samples showed that the radiometabolites of [^11^C]metoclopramide do not detectably cross the BBB. As a consequence, the PET signal in the brain predominantly consists in unchanged [^11^C]metoclopramide, which enables accurate quantification of P-gp function at the BBB using this radioligand [12]. Nevertheless, it cannot be excluded that a change in the metabolism of [^11^C]metoclopramide induced by disease state, genetic polymorphisms or drug-drug interactions (DDI) with modulators of cytochrome P450 (CYP) activity may impact the pharmacokinetics of [^11^C]metoclopramide. For instance, patients with neurological diseases are frequently treated with drugs that may precipitate DDIs through inhibition or induction of CYP activity. This is particularly the case for patients with drug-resistant epilepsy, bipolar disorders or severe depression, in whom CYP inducers such as phenytoin or carbamazepine are commonly prescribed [20,21]. It is crucial, therefore, to anticipate this situation and avoid biased estimation of P-gp function at the BBB in patients treated with modulators of CYP activity.

In the present study, the impact of the CYP inducer carbamazepine (CBZ) and the potent and broad spectrum CYP inhibitor ritonavir (RIT) on the brain and plasma kinetics of [^11^C]metoclopramide was investigated in rats. Kinetic modeling was performed to estimate the impact of these drugs on P-gp function at the BBB.

## 2. Materials and Methods

### 2.1. Treatments

CBZ (Tegretol^®^ oral suspension 20 mg/mL, 200 mL per bottle) was purchased from Novartis pharma laboratory, Basel, Switzerland. In animals of the CBZ group (*n* = 4), CBZ suspension was administered by oral gavage at a dose of 45 mg/kg, twice a day (8 A.M and 5 P.M) for 7 days before the PET imaging. Each animal received a total of 14 doses of CBZ before the PET experiment. Chronic administration of CBZ, at least 7 days, is required to induce multiple CYPs in rats [22,23].

RIT (Norvir^®^ powder for oral suspension, 100 mg per bag) was purchased from AbbVie laboratory, North Chicago, IL, USA. A RIT suspension (10 mg/mL) was freshly prepared prior to oral administration by adding 100 mg of RIT powder to 9.4 mL of water for injection. Animals of the RIT group (*n* = 4) received a single dose of 20 mg/kg of RIT 3 hours before the PET experiment. This acute administration protocol was shown to be well-tolerated and to inhibit multiple CYPs in rats [24,25].

Data obtained in the CBZ and the RIT groups were compared with data obtained in drug-naïve animals (baseline, *n* = 4)

### 2.2. Radiochemistry

[^11^C]metoclopramide was synthesized by automated radiomethylation using a TRACERlab^®^ FX C Pro module (GE Healthcare, Chicago, IL, USA) and cyclotron-produced [^11^C]CO_2_ according to a method described in the literature [26]. [^11^C]CO_2_ (50–70 GBq) was produced on a cyclone 18/9 cyclotron (18 MeV, IBA, Ottignies-Louvain-la-Neuve, Belgium) via the ^14^N(p, α)^11^C nuclear reaction by irradiation of an [^14^N]N_2_ target containing 0.15–0.5% of O_2_. [^11^C]CO_2_ was subsequently reduced to [^11^C]CH_4_ and iodinated to [^11^C]CH_3_I and finally converted into [^11^C]methyl triflate ([^11^C]CH_3_OTf). [^11^C]CH_3_OTf was bubbled into a solution of *O*-desmethyl-metoclopramide (ABX advanced biochemical compounds, Radeberg, Germany, 1 mg) and aqueous sodium hydroxide (3 M, 7 μL) in acetone (400 μL) at −20 °C for 3 min. The mixture was heated at 110 °C for 2 min, then the residual solvent was evaporated to dryness at 110 °C under vacuum for 30 s. Under cooling to 60 °C, a mixture of aqueous NaH_2_PO_4_ (20 mM)/CH_3_CN/H_3_PO_4_ (85/15/0.2 *v*/*v*/*v*) was added. Purification was realized by reversed phase high-performance liquid chromatography (HPLC, column: Waters Symmetry^®^ C18 7.8 × 300 mm, 7 μm) with a 501 HPLC Pump (Waters, Milford, CT, USA) using aqueous NaH_2_PO_4_ (20 mM)/CH_3_CN/H_3_PO_4_ (85/15/0.2 *v*/*v*/*v*, 5 mL/min) as eluent. UV detection (K2501, Knauer, Berlin, Germany) was performed at 220 nm. The product fraction was diluted with water (20 mL) and passed through a Sep-Pak^®^ C18 cartridge (Waters, Milford, CT, USA). The cartridge was rinsed with water (10 mL) and eluted with ethanol (2 mL). The eluate was diluted with saline (0.9% *w*/*v*, 8 mL) to afford ready-to-inject [^11^C]metoclopramide (1.9 ± 0.2 GBq) in 12 ± 3% radiochemical yield (RCY) within 40 min and with a molar activity (MA) of 88 ± 13 GBq/μmol at the end of bombardment (EOB) (*n* = 35).

### 2.3. Animals

Twenty-one Sprague Dawley rats (Janvier, Le Genest-Saint-Isle, France) were used for the study (mean weight: 277 ± 32 g). Animals were housed and acclimatized for at least 3 days before the experiments. Rats had free access to chow and water. All animal experiments were in accordance with the recommendations of the European Community (2010/63/UE) and the French National Committees (law 2013-118) for the care and use of laboratory animals. The experimental protocol was approved by a local ethics committee for animal use (CETEA) and by the French ministry of agriculture (APAFIS#746620161104 17049220 v2). The sample size for each group was based on previous studies with the aim to compare the brain distribution of another P-gp PET probe [27].

### 2.4. PET Acquisition

PET acquisitions were performed in Baseline, CBZ and RIT pretreated animals (*n* = 4 per condition) using an Inveon microPET scanner (Siemens, Knoxville, TN, USA). Anesthesia was induced and thereafter maintained using 3.5% and 1.5–2.5% isoflurane in O_2_, respectively. Dynamic 30-min PET scans were acquired starting with intravenous bolus injection of [^11^C]metoclopramide (34 ± 7 MBq, corresponding to 0.8 ± 0.2 µg of unlabeled metoclopramide via a catheter inserted in the caudal lateral vein.

### 2.5. Arterial Input Function

Repeated arterial blood sampling during PET acquisition is very challenging in rats. Arterial blood sampling, therefore, was conducted in separate groups of rats to measure the metabolism and arterial input function (AIF) of [^11^C]metoclopramide without (baseline) and with pretreatment with CBZ or RIT (*n* = 3 for each condition).

Blood samples (50 µL) were collected at designated time points from the femoral artery. Plasma was separated from whole blood by centrifugation (5 min, 2054 g, 4 °C) and 20 µL aliquots of plasma and whole blood were counted in a PET cross-calibrated gamma well counter (WIZARD2, PerkinElmer, Villebon-sur-Yvette, France). Radioactivity concentrations were corrected for radioactive decay from the injection time to obtain the whole-blood and plasma curves of total radioactivity (metabolites and parent). In addition, larger arterial blood samples (200 µL) were collected at designated time points. From these samples, 80 µL plasma was deproteinized with acetonitrile. The supernatant was injected into a HPLC system, equipped with an Atlantis^®^ T3 10 μm 10 × 250 mm column (Waters, Saint-Quentin-en-Yvelines, France) with a LB-514 radioactivity flow detector (Berthold, France, MX Z100 cell). The mobile phase was composed of water with 0.1% of trifluoroacetic acid (A) and acetonitrile with 0.1% of trifluoroacetic acid (B) delivered in a gradient elution mode at a flow rate of 5 mL/min. Solvent B increased linearly from 20 to 30% from 0 to 11 min. The [^11^C]metoclopramide parent fraction was calculated as a percentage of the total radioactivity (metabolites and parent).

For each animal, a 1-exponential decay function was fitted to the [^11^C]metoclopramide parent fraction over time, which was multiplied with the total radioactivity plasma curve to obtain a metabolite-corrected AIF. The average metabolite-corrected AIF of 3 animals per group was used for kinetic modeling.

### 2.6. PET Data Analysis

Images were reconstructed with the Fourier rebinning algorithm and the 3-dimensional ordered-subset expectation-maximization algorithm, including normalization, attenuation, scatter, and random corrections. Image analysis was performed using PMOD software (version 3.9, PMOD Technologies LLC, Fällanden, Switzerland). The brain and the blood pool (i.e., from the left ventricle of the heart) were manually delineated, as previously described, to generate corresponding time–activity curves (TACs) [12]. Radioactivity was corrected for ^11^C decay and expressed as standardized uptake value (SUV, unitless) after correction for injected dose and animal weight. SUV-normalized PET images were generated for visual comparison. Area under the brain curve between 0 and 30 min (AUC_brain_) was calculated to estimate brain exposure to [^11^C]metoclopramide.

Kinetic modeling was also performed using PMOD software. For each animal, the total volume of distribution (*V*_T_) of [^11^C]metoclopramide in the brain was first estimated with Logan plot analysis [12] using the average metabolite-corrected AIF obtained for each condition (baseline, CBZ or RIT). In addition, TACs obtained from the heart blood pool in each individual were used to generate an image-derived input function (IDIF). Each individual IDIF was corrected by a group-specific, average whole-blood/plasma radioactivity ratio which was obtained from the arterial blood samples collected in the separate groups of animals (see above). This ratio was 1.5 for the baseline group, 1.7 for the CBZ group and 1.6 for the RIT group. The IDIF was then corrected for radiometabolites using the corresponding parent [^11^C]metoclopramide fraction measured in plasma in each condition in the separate groups of animals (see above) and used to calculate *V*_T_.

Area under the plasma curve between 0 and 30 min was calculated using both methods (AUC_AIF_, *n* = 3 per condition, and AUC_IDIF_, *n* = 4 per condition).

### 2.7. Statistical Analysis

Statistical tests were performed using GraphPad Prism software (version 9.1.2, (GraphPad, La Jolla, CA, USA). The level of statistical significance was set to *p* ≤ 0.05. A one-factor ANOVA test was performed to compare the mean values of AUC_brain_, AUC_AIF_ and AUC_IDIF_ between the three groups, and a Tukey’s post hoc test was used for multiple comparisons. The values were considered independent of each other. Normal distribution of the data was verified by the Shapiro–Wilk test and homoscedasticity by a Bartlett’s test.

A two-factor ANOVA test was performed to compare the mean values of V_T,AIF_ and V_T,IDIF_ between the three groups, Tukey’s post hoc was used for multiple comparisons. The three groups of rats were independent. Normal distribution of the data was verified by the Kolmogorov–Smirnov test and homoscedasticity by a Levene’s test.

V_T,AIF_ and V_T,IDIF_ were correlated using a simple linear regression and the goodness of fit was estimated by the correlation coefficient R^2^.

## 3. Results

Visual comparison of SUV-normalized PET images displayed the typical brain distribution of [^11^C]metoclopramide, which crosses the BBB, even when P-gp is fully functional [12]. No difference could be observed between the baseline and CBZ-treated animals regarding the distribution of [^11^C]metoclopramide in the brain and in peripheral tissues. However, a higher PET signal was observed in RIT-treated animals, in both the brain and peripheral tissues, suggesting a substantial change in the kinetics of [^11^C]metoclopramide (Figure 1).

Whole brain TACs are shown in Figure 2A, confirming the visual observation from the PET images. At the end of the PET acquisition, brain uptake was not significantly different in baseline (SUV_30min_ = 0.263 ± 0.021) and CBZ-treated animals (SUV_30min_ = 0.233 ± 0.061). Brain uptake was higher in the RIT group (SUV_30min_ = 0.398 ± 0.041) compared with baseline (*p* < 0.01) and CBZ-treated rats (*p* < 0.01). Consistently, CBZ did not increase the overall brain exposure (AUC_brain_) to [^11^C]metoclopramide while RIT did (*p* < 0.05; Figure 2B).

Metabolism of [^11^C]metoclopramide was assessed by determining the fraction of parent (unmetabolized) [^11^C]metoclopramide in plasma over time (Figure 3). The fraction of parent [^11^C]metoclopramide was not significantly different between groups at 30 min after injection, which corresponds to the end of PET acquisition. However, the fraction of parent [^11^C]metoclopramide was significantly lower in the RIT group at 5 min after injection, but became similar to baseline and CBZ thereafter. Importantly, CBZ did not impact the fraction of parent [^11^C]metoclopramide in plasma compared with baseline at any time point (Figure 3).

The mean metabolite-corrected arterial input functions (AIF) obtained in each condition (*n* = 3) are shown in Figure 4A. The area under the TACs (AUC_AIF_) was not significantly different between conditions, although a trend towards decreased plasma exposure was observed in CBZ or RIT pretreated animals compared with baseline (Figure 4C). Similar results were obtained when comparing the metabolite-corrected IDIFs of [^11^C]metoclopramide in plasma, with a larger number of animals per group (*n* = 4, Figure 4B,C). The IDIF tended to overestimate the AIF. In all tested conditions, AUC_IDIF_ was approximately two-fold higher than AUC_AIF_. However, for both methods (AIF or IDIF), pretreatment with CBZ or RIT did not significantly impact the plasma exposure to [^11^C]metoclopramide (Figure 4C). AUC_,_ expressed in SUV.min, is normalized to the injected dose. Plasma AUC, therefore, is inversely correlated with the plasma clearance (Clearance = dose/AUC). It can be concluded, therefore, that CBZ and RIT did not impact the plasma clearance of [^11^C]metoclopramide in rats.

The brain distribution volume (*V*_T_), which corresponds to the brain/plasma concentration ratio at steady state [12], was estimated using the Logan graphical method to (i) compare the BBB penetration of [^11^C]metoclopramide between conditions and (ii) validate the IDIF-based method. There was a good correlation between *V*_T,IDIF_ and *V*_T,AIF_ (*p* < 0.001, R^2^ = 0.86). Neither *V*_T,AIF_ values nor *V*_T,IDIF_ values were significantly different between CBZ-treated and baseline rats. However, both *V*_T,AIF_ and *V*_T,IDIF_ values were significantly higher in the RIT group than in the baseline (*p* < 0.001) and the CBZ groups (*p* < 0.01), suggesting better passage of [^11^C]metoclopramide across the BBB in RIT-treated animals (Figure 5).

## 4. Discussion

The main objective of this preclinical study was to investigate the potential impact of co-administered drugs on the plasma and brain kinetics of [^11^C]metoclopramide, a PET tracer used to estimate P-gp function at the BBB.

A large body of translational research suggests that P-gp function at the BBB may provide a potential biomarker of drug-resistant epilepsy [11,28,29,30]. In contrast to previously developed radiotracers, it was shown that [^11^C]metoclopramide may be able to detect an induction of P-gp function at the BBB [16]. [^11^C]metoclopramide PET imaging may provide, therefore, a clinical tool to unveil and localize epileptic foci based on the local overexpression of P-gp at the BBB [11]. However, treatment with antiepileptic drugs (AEDs) cannot reasonably be interrupted before PET imaging, given the risk for increased seizure frequency and magnitude in patients. It is essential, therefore, to anticipate the impact of concomitant drug intake on the kinetics of [^11^C]metoclopramide, given the high propensity of many AEDs such as phenobarbital, phenytoin or CBZ to precipitate DDIs through induction of CYPs [31].

The anticonvulsant CBZ is widely prescribed in the long-term therapy of epilepsy and is a potent inducer of CYP3A4 and CYP2B6 [32]. It was shown that chronic treatment, at least 7 days, is required to achieve induction of multiple CYPs in Sprague Dawley rats [23]. The dose and administration scheme of CBZ that we selected was shown to be well-tolerated, induce CYPs and increase the plasma clearance of the CYP3A substrate rivaroxaban in Wistar rats [22]. In the present study, determination of effective induction of CYPs in each individual rat has not been performed. However, our results showed that 7 days chronic CBZ treatment did not impact the plasma or the brain kinetics of [^11^C]metoclopramide in rats. Moreover, the brain penetration of [^11^C]metoclopramide, estimated as *V*_T_, was not changed by chronic CBZ treatment. First, this shows that [^11^C]metoclopramide is poorly vulnerable to DDIs precipitated by CYP inducers in rats. This may not be the case in humans as oral administration of rifampicin, another major CYP inducer, led to a significant decrease in the plasma concentration of unlabeled metoclopramide in healthy volunteers [33]. However, this result suggests that CBZ does not substantially induce or inhibit P-gp function at the BBB in healthy Sprague Dawley rats.

Enzymes of the CYP3A subfamily, notably CYP3A4, have the greatest abundance in the liver and intestine, and are responsible for the metabolism of a large number of small-molecule drugs, including PET tracers [19]. Many studies have shown that members of the CYP3A subfamily are inducible enzymes and the mechanism for induction has been thoroughly explored [34]. As a consequence, most induction DDIs are related to CYP3A enzymes and the risk for such DDIs should be explored during drug development [35]. In comparison, CYP2D6 does not appear to be inducible [31] and investigation of CYP2D6 induction is not recommended during drug development [35]. In vitro studies using human enzymes concluded that metoclopramide is predominantly metabolized by CYP2D6 in humans [36]. From an imaging perspective, the negligible impact of enzyme induction with CBZ on the plasma kinetics of [^11^C]metoclopramide is a major asset compared with previous P-gp probes such as [^11^C]verapamil, a CYP3A4 substrate which was shown to be vulnerable to CYP induction [37,38]. In addition, radiometabolites of [^11^C]verapamil were shown to cross the BBB in rats, thereby potentially confounding the measurement of cerebral P-gp function with this PET tracer [37]. Against this background, [^11^C]*N*-desmethyl-loperamide has been specifically designed as a metabolically-stable PET tracer to study P-gp function at the human BBB [39]. However, this PET probe shows extensive peripheral metabolism and brain uptake of radiolabeled metabolites in mice [40]. Moreover, the very low baseline brain uptake of [^11^C]*N*-desmethyl-loperamide does not allow for detection of an induction of P-gp at the BBB. Moreover, [^11^C]*N*-desmethyl-loperamide may lack the sensitivity to detect moderate changes in P-gp function compared with other radioligands [14]. Altogether, our preclinical results support the applicability of [^11^C]metoclopramide as a PET probe in situations in which concomitant therapy with CYP inducers cannot be avoided. The impact of CYP induction on the metabolism of newly proposed ^18^F-labeled PET probes such as [^18^F]MC225 has not been reported to our knowledge [41].

Data regarding the impact of concomitant drugs on the peripheral metabolism of [^11^C]metoclopramide have been reported. In rats, co-injection of a pharmacological dose of metoclopramide (3 mg/kg, i.v) with [^11^C]metoclopramide was shown to increase the parent fraction of [^11^C]metoclopramide in plasma [12]. This non-linearity in radiotracer metabolism was attributed to a saturable uptake transport of [^11^C]metoclopramide by the liver, probably mediated by organic cation transporters (OCT1/2) in rodents [26,42]. However, non-linearity was not observed in humans in whom co-injection of unlabeled metoclopramide (10 mg, i.v) did not impact the plasma kinetics of [^11^C]metoclopramide [15]. Compared with rodents (fraction of parent [^11^C]metoclopramide ~10% at 30 min) [12,26], or non-human primates (parent [^11^C]metoclopramide was barely detectable after 30 min) [43], a microdose of [^11^C]metoclopramide is considerably less metabolized in humans (fraction of parent [^11^C]metoclopramide ~60% at 30 min) [13]. This suggests significant species differences in the metabolism of [^11^C]metoclopramide. Consistently, the rat ortholog of human CYP2D6 comprises six CYP2D isozymes: (CYP2D15 and CYP2D18) and notable differences have been reported in the enzymatic activity between the rat and the human isoforms [44]. The negligible impact of CBZ on the peripheral metabolism of [^11^C]metoclopramide suggests that inducible enzymes such as CYP3A are poorly involved in the metabolism of metoclopramide in rats. In humans, the impact of genetic polymorphism of the *CYP2D6* gene on the plasma kinetics of [^11^C]metoclopramide remains to be investigated [45]. However, in humans, it is possible to perform arterial blood sampling during brain PET acquisition and estimate P-gp function using pharmacokinetic modeling which takes into account any change in the peripheral metabolism of [^11^C]metoclopramide [13].

We next investigated the impact of CYP inhibition on the plasma and brain kinetics of [^11^C]metoclopramide. There is no validated broad spectrum CYP inhibitor among AEDs that can precipitate DDI in rats. Therefore, we selected RIT, which is a potent and well-characterized CYP inhibitor, including CYP3A and CYP2D6 [46]. The selected dose and administration scheme of RIT was shown to inhibit the metabolism of a CYP3A substrate in rats [24,25]. RIT did not impact the plasma clearance of parent [^11^C]metoclopramide, suggesting a limited vulnerability of [^11^C]metoclopramide to CYP inhibition-mediated DDIs. However, RIT increased the brain exposure and BBB penetration of [^11^C]metoclopramide compared with baseline or CBZ-treated rats (Figure 2 and Figure 5). RIT is a potent P-gp inhibitor in vitro [47] and at the intestinal level [46]. We first hypothesized that RIT may inhibit P-gp function at the BBB, thus increasing the brain penetration and exposure to [^11^C]metoclopramide. However, compared with the baseline or the CBZ group, RIT also notably increased the PET signal in tissues surrounding the brain, suggesting a global increase in the tissue distribution of [^11^C]metoclopramide. Moreover, enhanced diffusion of parent (unmetabolized) [^11^C]metoclopramide to tissues may decrease the proportion of unchanged [^11^C]metoclopramide in the circulation, assuming a limited and constant tissue distribution of radiometabolites. This is consistent with the lower percentage of parent [^11^C]metoclopramide in plasma observed at 5 min in the RIT group. However, an increase in the distribution of [^11^C]metoclopramide in other tissues than the brain was not observed using the potent P-gp inhibitor tariquidar, which specifically increased the brain uptake of [^11^C]metoclopramide, with a limited impact on surrounding tissues [12,14]. Further experiments, therefore, are needed to untangle the mechanisms involved in the increased brain uptake of [^11^C]metoclopramide precipitated by RIT or to validate RIT as an inhibitor of P-gp at the BBB.

## 5. Conclusions

The present study shows that CYP induction with CBZ as well as CYP inhibition with RIT have a negligible impact on the plasma kinetics of [^11^C]metoclopramide in rats. This supports the use of [^11^C]metoclopramide as a PET probe for P-gp function at the BBB in situations in which concomitant drug treatment cannot be avoided. CBZ did also not impact the brain kinetics and BBB penetration of [^11^C]metoclopramide, thus confirming that estimation of P-gp function at the BBB is not biased by this model CYP inducer. RIT increased the brain exposure to [^11^C]metoclopramide, thus apparently behaving as an inhibitor of P-gp at the BBB. Further experiments using complementary approaches, however, are needed to confirm this mechanism.

## Figures and Tables

**Figure 1 pharmaceutics-14-02650-f001:**
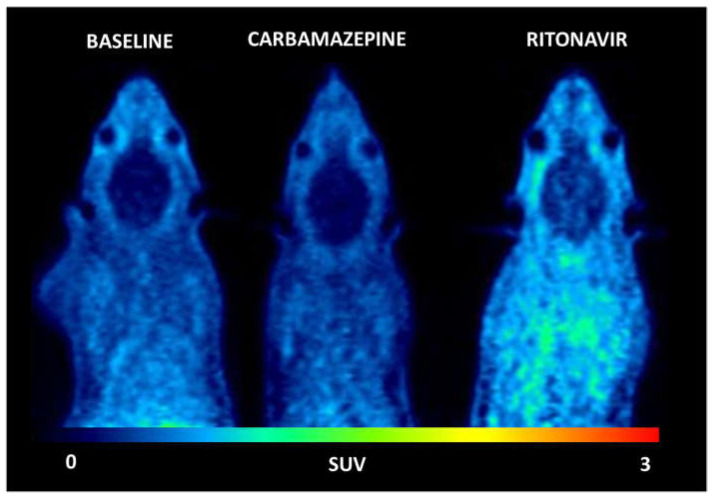
Representative PET summation images (from 0 to 30 min) of [^11^C]metoclopramide from one baseline rat and one rat pretreated with carbamazepine (45 mg/kg twice a day for 7 days before PET) or ritonavir (20 mg/kg, 3 h before PET).

**Figure 2 pharmaceutics-14-02650-f002:**
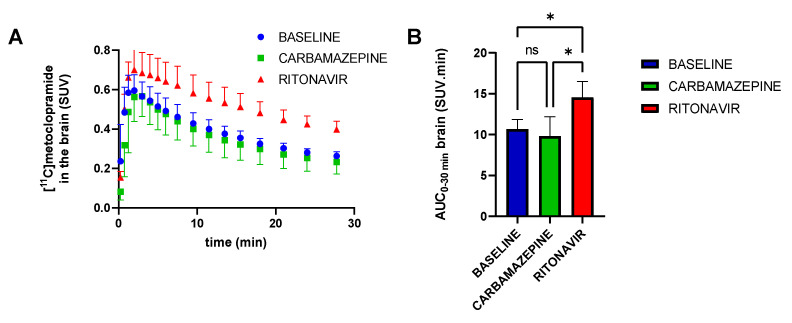
Whole-brain time–activity curves (**A**) and area under the whole brain curve from 0 to 30 min (AUC_0–30 min_, (**B**) for groups without (baseline) and with pretreatment with carbamazepine or ritonavir. Data are reported as mean ± S.D with *n* = 4 per condition. ns = not significant, * *p* < 0.05 (one-factor ANOVA and Tukey’s post hoc test for multiple comparisons).

**Figure 3 pharmaceutics-14-02650-f003:**
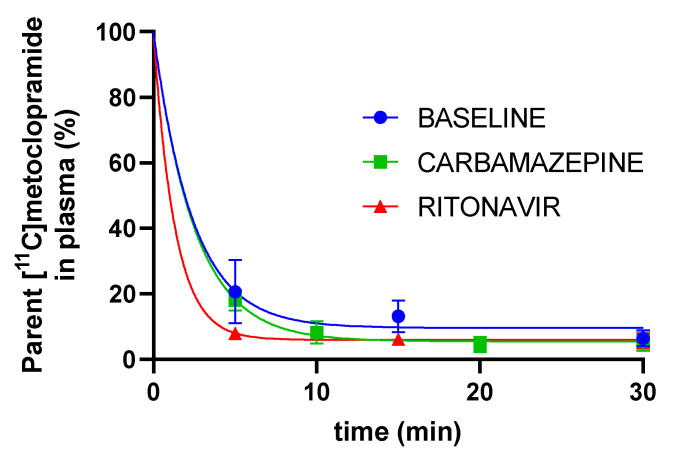
Fraction of parent (unmetabolized) [^11^C]metoclopramide in plasma over time without and with pretreatment with carbamazepine or ritonavir (mean ± S.D with *n* = 3 per condition).

**Figure 4 pharmaceutics-14-02650-f004:**
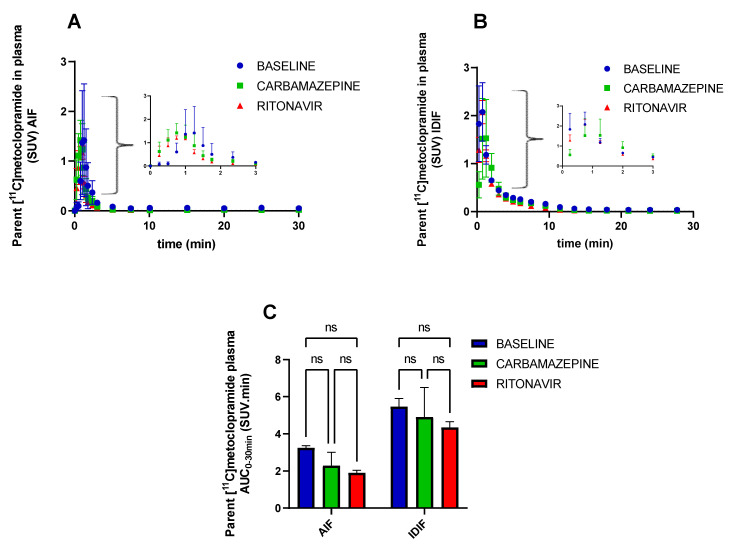
Concentration of parent (unmetabolized) [^11^C]metoclopramide in plasma over time determined from the sampled arterial input function (AIF, (**A**) or from the image-derived input function (IDIF, (**B**). In (**C**), the area under the curve from 0 to 30 min (AUC_0–30 min_) of parent [^11^C]metoclopramide in plasma is shown. Data are reported as mean ± S.D with *n* = 3 (AIF) or *n* = 4 (IDIF) per condition. ns = not significant (two-way ANOVA with Tukey’s post hoc test for multiple comparisons).

**Figure 5 pharmaceutics-14-02650-f005:**
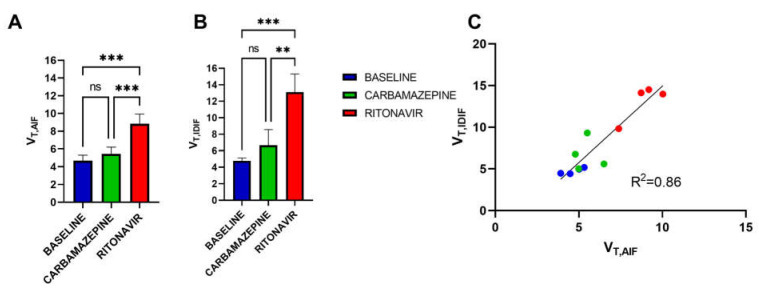
Brain volume of distribution (*V*_T_) of [^11^C]metoclopramide in groups without (baseline) and with pretreatment with carbamazepine or ritonavir determined with the arterial input function (*V*_T,AIF_, (**A**) or the image-derived input function (*V*_T,IDIF_, (**B**) and linear correlation between *V*_T,IDIF_ and *V*_T,AIF_ (C). Data are reported as mean ± S.D with *n* = 3 (AIF) or *n* = 4 (IDIF) per condition. *** *p* < 0.001, ** *p* < 0.01, ns = not significant (two-way ANOVA with Tukey’s post hoc test for multiple comparison).

## Data Availability

Data are freely available upon request to the corresponding author.

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
