# Peer review of "Impact of Cytochrome Induction or Inhibition on the Plasma and Brain Kinetics of [11C]metoclopramide, a PET Probe for P-Glycoprotein Function at the Blood-Brain Barrier"

_pharmaceutics, 2022, doi:10.3390/pharmaceutics14122650_

Round 1

Reviewer 1 Report

The study is very interesting (accessing brain kinetics of drugs using PET), pertinent, and of interest to the readers of this journal.

The availability of data by the authors reflects the clarity and openness desirable in science. We congratulate the authors on this.

The methodology is relevant and the experimental design is well-structured. Furthermore, this article has a good statistical methodology in the data analysis (supported by several tests), and it is clearly described. It should be noted that the authors checked in advance the normal distribution of the data.

The specific comments are stated below:

Comment #1- We found cited references excessive (60 references!!!) and there are too many “self-citations”.

Comment #2- Seem to have redundancy in the phrases contained in lines 212-213. From what we understand in the text, the effect of the variable is measured in the experimental unit, which in this case seems to be each animal group containing several sample units, therefore, any estimate of parameters between groups seems to be independent.

Comment #3- Lines 207-208: We noticed a different number of observations for AUC variable (line 202). Was weighting done to the data when the Tukey test (multiple comparisons) was attempted? [Note that one of the disadvantages of Tukey test (compared to LSD) is the more incorrect conclusions about similar means under hypothesis (type II errors). [Biostatistics For Animal Science by William R Lamberson e Miroslav Kaps ISBN: 9781786390356Editor: Cabi Publishing].

Author Response

The study is very interesting (accessing brain kinetics of drugs using PET), pertinent, and of interest to the readers of this journal. The availability of data by the authors reflects the clarity and openness desirable in science. We congratulate the authors on this.

The methodology is relevant and the experimental design is well-structured. Furthermore, this article has a good statistical methodology in the data analysis (supported by several tests), and it is clearly described. It should be noted that the authors checked in advance the normal distribution of the data.

We thank the reviewer for his/her comments.

The specific comments are stated below:

Comment #1- We found cited references excessive (60 references!!!) and there are too many “self-citations”.

We have noticed that several references were cited twice in the initial manuscript due to an error with our citation software. This is now corrected. Moreover, we have reduced the total number or citation down to 47 with particular attention to limit self-citation. Please note that self-citation is difficult to avoid because 11C-metoclopramide has been developed by the teams in University Paris-Saclay and the University of Vienna, who paved the way for the present collaborative study.

Comment #2- Seem to have redundancy in the phrases contained in lines 212-213. From what we understand in the text, the effect of the variable is measured in the experimental unit, which in this case seems to be each animal group containing several sample units, therefore, any estimate of parameters between groups seems to be independent.

Thank you. The repeated information has been removed (line 228).

Comment #3- Lines 207-208: We noticed a different number of observations for AUC variable (line 202). Was weighting done to the data when the Tukey test (multiple comparisons) was attempted? [Note that one of the disadvantages of Tukey test (compared to LSD) is the more incorrect conclusions about similar means under hypothesis (type II errors). [Biostatistics For Animal Science by William R Lamberson e Miroslav Kaps ISBN: 9781786390356Editor: Cabi Publishing].

We confirm that the number of observations is different for AUCAIF (n=3, derived from invasive experiments in rats, line 172) and AUCIDIF (n=4, derived from PET images, line 161-162). We fully agree with the reviewer and did not accordingly compare AUCAIF values with AUCIDIF values. Only AUC values obtained using the same method (IDIF or AIF) have been compared between groups (baseline, RIT and CBZ), with the same number of observations in each group.

Reviewer 2 Report

This article estimates using [11C]metoclopramide as a PET probe for the function of P-gp in the BBB in situations where concomitant drug treatment cannot be avoided.

The authors did not observe any impact of carbamazepine (CBZ) on the fraction of the parent [11C]metoclopramide in plasma compared to baseline at any time. Furthermore, the brain distribution volume (VT) was not significantly different between CBZ-treated and baseline rats. Carbamazepine is a substrate and inducer of cytochrome P450 (CYP) 2B1/CYP2B2 and CYP3A in rats. However, some strain differences are observed in CYP3A induction in Wistar (WI) and Sprague-Dawley (SD) rats. CYP3A2 mRNA was more strongly induced by dexamethasone, a typical inducer of CYP3A, together with CYP3A1 mRNA, in WI rats than in SD (by twofold) (Kishida T, Muto S, Hayashi M, Tsutsui M, Tanaka S, Murakami M, Kuroda J. Strain differences in the expression of liver cytochrome P450 1A and 3A between Sprague-Dawley and Wistar rats. J Toxicol Sci. 2008 Oct;33(4):447-57. doi: 10.2131/jts.33.447. PMID: 18827444.)

How do the authors know they have the correct dose of carbamazepine (that is, the dose that can induce CYP3A in SD rats)? The reference in the methodology is for Wistar rats, while the authors work with SD rats.

Why was the study conducted on a small number of rats (only 4 per group)?

[11C] metoclopramide is an interesting PET probe for the P-gp function in the BBB. However, metoclopramide is metabolized by CYP2D6 and is a reversible inhibitor of CYP2D6. CYP2D6 is not an inducible CYP isoform in humans. However, the frequency of ultra-fast metabolism (UM) phenomena has been reported in up to 10% of Caucasians (Europe, North America) and is sometimes higher in some groups, such as Ashkenazi. Therefore, a change in the metabolism of [11C] metoclopramide induced by genetic polymorphisms associated with CYP2D6 can affect the pharmacokinetics of [11C]metoclopramide and, finally, the PET imaging of [11C]metoclopramide. Therefore, although the study of induction of CYP2D6 is not recommended during drug development, it should be considered when introducing this substance into the clinic. Currently, [11C] metoclopramide can be only an interesting PET probe for the function of P-gp in the BBB in rats.

Author Response

We gratefully thank the reviewer for his/her comments.

This article estimates using [11C]metoclopramide as a PET probe for the function of P-gp in the BBB in situations where concomitant drug treatment cannot be avoided.

The authors did not observe any impact of carbamazepine (CBZ) on the fraction of the parent [11C]metoclopramide in plasma compared to baseline at any time. Furthermore, the brain distribution volume (VT) was not significantly different between CBZ-treated and baseline rats. Carbamazepine is a substrate and inducer of cytochrome P450 (CYP) 2B1/CYP2B2 and CYP3A in rats. However, some strain differences are observed in CYP3A induction in Wistar (WI) and Sprague-Dawley (SD) rats. CYP3A2 mRNA was more strongly induced by dexamethasone, a typical inducer of CYP3A, together with CYP3A1 mRNA, in WI rats than in SD (by twofold) (Kishida T, Muto S, Hayashi M, Tsutsui M, Tanaka S, Murakami M, Kuroda J. Strain differences in the expression of liver cytochrome P450 1A and 3A between Sprague-Dawley and Wistar rats. J Toxicol Sci. 2008 Oct;33(4):447-57. doi: 10.2131/jts.33.447. PMID: 18827444.)

How do the authors know they have the correct dose of carbamazepine (that is, the dose that can induce CYP3A in SD rats)? The reference in the methodology is for Wistar rats, while the authors work with SD rats.

We thank the reviewer for this very important comment.

The dose of carbamazepine used to induce CYP (45 mg/kg x 2 daily) has been selected based on two different studies performed in Wistar (Ngo et al., 2020, (45 mg/kg x 2 daily) or in Sprague-Dawley rats (Tateishi et al., 1999, 60 or 100 mg/kg daily).

Thanks to the reviewer’s comment, we noticed that the Tateishi study was cited in the discussion but not in the Material and Method section, where the carbamazepine dose is justified. This is now corrected, please see line 110 (refs 22 and 23).

Please note that the maximal dose of 100 mg/kg daily described by Tateishi in Sprague-Dawley rats was poorly tolerated in our hands and did not allow for the safe and prolonged anesthesia required for PET acquisition. We therefore used the dose proposed by Ngo et al (45 mg/kg x 2/j).

Moreover, we have clarified the strain of rats used in the cited references lines 325 and 328 in the discussion.

To highlight reviewer’s comment and acknowledge the limitation of our study, we have added the following sentence line 328-329 : “In the present study, determination of effective induction of CYPs in each individual rat has not been performed.”

For reviewer information, drug-drug interaction involving carbamazepine as a potential inducer of CYP was a major concern for our lab because we are starting a program in patients with drug-resistant epilepsy. We therefore preferred using carbamazepine rather than dexamethasone to estimate the impact of this particular drug on the PK of 11C-metoclopramide.

Why was the study conducted on a small number of rats (only 4 per group)?

PET imaging is indeed a “low-throughput” and technically demanding method compared with in vitro and ex vivo PK studies used in the field of transporters and metabolism. PK PET studies in rats focusing on transporters at the BBB are usually performed using similar sample size (3-5 per group). This is generally sufficient to highlight differences of relevance for PK, as highlighted by the significant increase in VT observed using ritonavir. An additional reference has been added line 157-158 to justify the use of 4 animals per condition.

[11C]metoclopramide is an interesting PET probe for the P-gp function in the BBB. However, metoclopramide is metabolized by CYP2D6 and is a reversible inhibitor of CYP2D6. CYP2D6 is not an inducible CYP isoform in humans. However, the frequency of ultra-fast metabolism (UM) phenomena has been reported in up to 10% of Caucasians (Europe, North America) and is sometimes higher in some groups, such as Ashkenazi. Therefore, a change in the metabolism of [11C] metoclopramide induced by genetic polymorphisms associated with CYP2D6 can affect the pharmacokinetics of [11C]metoclopramide and, finally, the PET imaging of [11C]metoclopramide. Therefore, although the study of induction of CYP2D6 is not recommended during drug development, it should be considered when introducing this substance into the clinic. Currently, [11C] metoclopramide can be only an interesting PET probe for the function of P-gp in the BBB in rats.

We thank the reviewer for his/her comment. We have added a paragraph to address this point line 382-386.

Reviewer 3 Report

The studies entitled: “Impact of cytochrome induction or inhibition on the plasma and brain kinetics of [11C]metoclopramide, a PET probe for P-glycoprotein function at the blood-brain barrier “ (Manuscript ID: pharmaceutics-2058870) are very interesting and certainly experimentally demanding. I have a few comments on the manuscript:

1.      Line 248-252. Why the fraction of parent [11C]metoclopramide was significantly lower in the RIT group at 5 min after injection?

2.      The parent [11C]metoclopramide in the RIT group was practically the same to the CBZ group after 5 min. While both group are similar to baseline.   3.      What kind of labeled radiolabeled metabolites are formed? They should be determined, e.g. by spectroscopic methods. Have such studies been conducted? Is there any information about this?   4.      It is known that radioisotopes affect the PET signal (e.g. [11C]Verapamil). Can the effect of metabolites that contribute to the PET signal be neglected?

Author Response

We gratefully thank the reviewer for his/her comments.

The studies entitled: “Impact of cytochrome induction or inhibition on the plasma and brain kinetics of [11C]metoclopramide, a PET probe for P-glycoprotein function at the blood-brain barrier “ (Manuscript ID: pharmaceutics-2058870) are very interesting and certainly experimentally demanding. I have a few comments on the manuscript:

  1. Line 248-252. Why the fraction of parent [11C]metoclopramide was significantly lower in the RIT group at 5 min after injection?

This is a very important comment. We have clarified the discussion line 400-405 to address this point.

  1. The parent [11C]metoclopramide in the RIT group was practically the same to the CBZ group after 5 min. While both group are similar to baseline.

We confirm that the proportion of parent [11C]metoclopramide is similar for the 3 groups (Baseline, CBZ and RIT) after 5 min.

  1. What kind of labeled radiolabeled metabolites are formed? They should be determined, e.g. by spectroscopic methods. Have such studies been conducted? Is there any information about this?

We thank the reviewer for this comment. We have clarified the introduction line 75-82 to address this point.

  1. It is known that radioisotopes affect the PET signal (e.g. [11C]Verapamil). Can the effect of metabolites that contribute to the PET signal be neglected?

Absolutely, the radiometabolites of [11C]Verapamil are known to account for the brain PET signal, which is mentioned lines 350-352, which impairs accurate quantification of P-gp function at the BBB. The negligible passage of the radiometabolites of  [11C]metoclopramide is a major advantage of over [11C]verapamil for PET quantification of P-gp function.